# Advances in Platelet-Rich Plasma Treatment for Spinal Diseases: A Systematic Review

**DOI:** 10.3390/ijms24087677

**Published:** 2023-04-21

**Authors:** Soya Kawabata, Koji Akeda, Junichi Yamada, Norihiko Takegami, Tatsuhiko Fujiwara, Nobuyuki Fujita, Akihiro Sudo

**Affiliations:** 1Department of Orthopaedic Surgery, School of Medicine, Fujita Health University, 1-98 Dengakugakubo, Kutsukake-cho, Toyoake 470-1192, Japan; soya.kawabata@fujita-hu.ac.jp (S.K.); nfujita@fujita-hu.ac.jp (N.F.); 2Department of Orthopaedic Surgery, Mie University Graduate School of Medicine, Tsu 514-8507, Japan; yamada-j@med.mie-u.ac.jp (J.Y.); n-takegami@med.mie-u.ac.jp (N.T.); tatsuhiko-f@med.mie-u.ac.jp (T.F.); a-sudou@med.mie-u.ac.jp (A.S.)

**Keywords:** spinal diseases, platelet-rich plasma, intervertebral disc degeneration, spinal fusion surgery, low back pain

## Abstract

Spinal diseases are commonly associated with pain and neurological symptoms, which negatively impact patients’ quality of life. Platelet-rich plasma (PRP) is an autologous source of multiple growth factors and cytokines, with the potential to promote tissue regeneration. Recently, PRP has been widely used for the treatment of musculoskeletal diseases, including spinal diseases, in clinics. Given the increasing popularity of PRP therapy, this article examines the current literature for basic research and emerging clinical applications of this therapy for treating spinal diseases. First, we review in vitro and in vivo studies, evaluating the potential of PRP in repairing intervertebral disc degeneration, promoting bone union in spinal fusion surgeries, and aiding in neurological recovery from spinal cord injury. Second, we address the clinical applications of PRP in treating degenerative spinal disease, including its analgesic effect on low back pain and radicular pain, as well as accelerating bone union during spinal fusion surgery. Basic research demonstrates the promising regenerative potential of PRP, and clinical studies have reported on the safety and efficacy of PRP therapy for treating several spinal diseases. Nevertheless, further high-quality randomized controlled trials would be required to establish clinical evidence of PRP therapy.

## 1. Introduction

Spinal diseases, including spinal degenerative diseases, the ossification of spinal ligaments, spinal deformities, and spinal cord injury (SCI) cause pain and neurological symptoms. These greatly affect patients’ activity of daily life (ADL) and quality of life (QOL). Low back pain (LBP) is one of the most common complaints of patients with spinal diseases. Disorders of the intervertebral disc, facet joint, sacroiliac arthritis, and lumbar nerve root can cause LBP, which often becomes chronic and intractable. These disorders are generally treated with medications and rehabilitation, but often with limited efficacy [1]. The development of new, more effective treatments for chronic LBP is desirable.

Lumbar spinal stenosis and lumbar degenerative spondylolisthesis can cause pain, numbness, and muscle weakness in the lower extremities. Conservative treatments, such as medication and rehabilitation, have a certain degree of effectiveness [2]. When conservative treatments are less effective, surgical treatment is recommended; however, the further development of conservative treatments is desirable. In the surgical setting, spinal fusion is often indicated in cases of high instability or degeneration. Spinal fusion is often performed in conjunction with bone grafting, but the grafted bone may fail to fuse, resulting in a pseudoarthrosis. The incidence of pseudoarthrosis in long instrumented posterior spinal fusion for adult spinal deformities is estimated to be from 25 to 35% [3], and the establishment of adjuvant therapies to increase the rate of bone fusion is needed. Thus, there are many areas in the field of spinal diseases where new treatment methods are desired.

Biological and/or cellular therapies have been utilized in a variety of regenerative medicine treatments [4,5,6,7]. Platelet-rich plasma (PRP) has been used clinically for tissue regeneration and repair [8]. In recent years, especially in the field of orthopedics, the regenerative capabilities of PRP have been shown to repair damaged tissues, such as tendons, ligaments, and cartilage [9,10]. Recently, a number of studies have reported on the treatment of spinal diseases with PRP [11]. However, the efficacy of PRP used in clinical applications is at times controversial due to the lack of quality clinical evidence.

Given the increasing popularity of PRP therapy, this article reviews basic research and emerging clinical applications in the current literature on treating spinal diseases with PRP. We first describe the biology of PRP and discuss the classification of PRP. Next, we review the literature on the use of PRP for treating spinal disorders and divide the research into basic and clinical studies. We review the clinical studies separately as follows: clinical application for intradiscal therapy, spinal fusion surgery, intraarticular therapy for the facet joint or sacroiliac joint pain, and epidural therapy.

## 2. Biology of Platelets

### 2.1. Platelet Activation and Secretion

Upon vessel injury, circulating platelets are exposed to the vascular wall and soluble agonists, which induce platelet activation, leading to clot formation. Platelets contain several types of secretory inclusions, such as dense granules, α-granules, and lysosomes [12]. Among them, α-granules are the most abundant, with approximately 50–80 granules per platelet, ranging in size from 200 to 500 nm. α-granules contain membrane-bound and soluble proteins. Membrane-bound proteins include integrins, immunoglobulin family receptors, and leucine-rich repeat family receptors [13]. Following platelet activation, membrane-bound proteins are expressed on the platelet surface, whereas soluble proteins are released into the extracellular compartment. Importantly, α-granules contain small vesicles called exosomes [14], which can also be released following platelet activation. Proteomic studies revealed that more than 300 soluble proteins are released from activated α-granules [15,16]. These bioactive proteins released from α-granules play diverse roles in hemostasis, inflammation, antimicrobial host defense, angiogenesis, and wound healing [12]. Specific examples of these proteins are shown in Table 1.

Platelets contain 3–8 dense granules, which contain high concentrations of cations (Ca^2+^, Mg^2+^, K^+^), phosphates, bioactive amines (serotonin, histamine), and nucleotides (ADP, ATP, cAMP, etc.) [12]. Only one to three lysosomes are present per platelet. Lysosomes contain enzymes involved in the degradation of proteins, carbohydrates, and lipids.

### 2.2. Platelet Extracellular Vesicles

Extracellular vesicles (EVs) include exosomes (30–100 nm in diameter) and micro-vesicles ([MV] 100–1000 nm in diameter). EVs contain membrane proteins, messenger RNA (mRNA), microRNA (miRNA), long non-coding RNA (lncRNA), and circular RNA (circRNA); they are generated and released from the vast majority of cell types into the extracellular space [17]. It has been reported that EVs are also released from activated platelets [18,19], and play an essential role in coagulation, the immune response, inflammation, angiogenesis, and tissue repair [20].

### 2.3. PRP

PRP is a fraction of plasma with a high concentration of platelets obtained by centrifugation. Theoretically, PRP with supra-physiological concentrations of bioactive proteins, including platelet EVs, have the potential to stimulate regenerative and/or reparative effects in the target tissues and/or organs. In particular, PRP has been used in clinical settings for repairing tissues in the musculoskeletal system, including bone, cartilage, intervertebral disc, tendons, joints, and in the nervous system [21].

## 3. PRP Classification

There are a wide variety of methods used in the purification of PRP; depending on the centrifugal conditions and extraction method, the concentration of platelets, white blood cells, and growth factors varies. Additionally, there are many commercially available kits that aim to efficiently purify highly stable PRP, but the quality of the purified PRP varies depending on the kit used. This is one of the obstacles for increasing the efficacy of PRP therapy.

There are two main PRP purification methods: the open and closed techniques. In the open technique, the blood is in contact with the environment in the working area during PRP purification. Pipettes and tubes are sterilized separately and used in the PRP purification process. In contrast, the closed technique uses commercially available equipment and kits, and the blood and PRP are not exposed to the environment during the preparation process [22]. The open technique has the advantage of being low cost, but there is a risk of bacterial contamination. The closed technique has a lower contamination risk, but is more costly; additionally, certain kits provide lower yield in terms of platelet concentration. Kushida et al. [23] compared seven systems and evaluated the purified PRP in detail. Centrifugation was performed two times in four of the systems and once in three of the systems, each system following the original protocol for PRP preparation. PRP was separated by tube centrifugation in four systems, gel separation in two systems, and fully automated centrifugation in one system. The required whole blood volume ranged from 8 to 60 mL, the final volume of PRP ranged from 0.6 to 3 mL, and the average platelet concentration of PRP varied widely from 8.8 × 10^4^/µL to 152.1 × 10^4^/µL, depending on the system. Although PRP containing more than a specific concentration of platelets tends to have higher concentrations of platelet-derived growth factor-AB (PDGF-AB), the relationship was not always directly proportional. The concentrations of transforming growth factor beta-1 (TGF-β1) and vascular endothelial growth factor (VEGF) vary widely from system to system. Platelet concentration ratios from less than 2-fold to an 8.5-fold increase have been reported over baseline; however, reports suggest a 3- to 5-fold increase is desirable [23,24]. A certain concentration of platelets is necessary because a low platelet concentration tends to reduce the number of growth factors.

As mentioned above, the content of PRP is considered to have a significant impact on treatment efficacy, and evaluating the content and quality of PRP is essential to validate its efficacy. DeLong et al. proposed a classification system based on Platelet concentration, Activation, or not, and leukocyte (White blood cell) concentration (PAW classification), which can be used to quickly evaluate the PRP preparations used in multiple studies and clinical practice [25].

The activation of PRP namely refers to two main processes: degranulation and cleavage-released growth factors from platelets. This process turns liquid plasma into a solid clot or membrane [26]. Exogenous activation techniques of PRP have been used for in vivo and clinical studies. PRP is usually activated by the addition of calcium chloride and/or thrombin, freezing and thawing, or exposure to collagen [11]. In a systematic review and meta-analysis, activated PRP was reported to be more effective for improving pain and functionality in patients with knee OA compared with non-activated PRP [27]. Additionally, Gentile reported that non-activated PRP was more useful for hair growth than activated PRP [28]. When PRP is injected into soft tissue, activation prior to administration is not always necessary because natural collagen type I acts as the activator [28].

Various basic and clinical studies have reported on the role of leukocyte content in the efficacy of PRP, but no consensus has been reached [29]. High concentrations of leukocytes may negatively affect PRP therapy, as leukocytes (especially neutrophils) act as inflammatory mediators. Nevertheless, leukocytes play an important role in the wound healing process, and their bactericidal activity has been reported to be beneficial for the treatment of bedsores and extensive soft tissue injuries [25]. Jia et al. reported that the presence of leukocytes in PRP may stimulate an inflammatory response at the cellular level [30]. Yan et al. reported that Leukocyte-poor PRP (Lp-PRP) significantly induced tendon regeneration compared to Leukocyte-rich PRP (Lr-PRP) in animal studies [31]. The results of clinical trials on patellar tendonitis [32], Achilles tendinopathy [33], and lateral epicondylitis [34] suggest that there was no difference in treatment outcomes between the Lr-PRP and Lp-PRP groups.

Dohan et al. used a simpler classification: Pure Platelet-Rich Plasma (P-PRP), Leukocyte-and Platelet-Rich Plasma (L-PRP), and Pure Platelet-Rich Fibrin (P-PRF), depending on whether the preparations were plasma or fibrin products and whether they contained white blood cells [35]. PRF is purified by collecting blood in dry glass or glass-coated plastic tubes and immediately centrifuging it at a low RPM. PRF preparations have a high-density fibrin network, meaning they can be handled as if they are a solid material [36]. Mishra et al. proposed dividing PRP preparations into eight categories based on white blood cell (WBC) count, activation or lack of, and platelet concentration (subtype), as follows [37]. Type 1: Increased WBCs without activation; Type 2: Increased WBCs with activation; Type 3: Minimal or no WBCs without activation; Type 4: Minimal or no WBCs with activation. Subtype A contains an increased platelet concentration at or above five times the baseline. Subtype B contains an increased platelet concentration less than five times the baseline. This classification is simple and best reflects the characteristics of PRP. The present review uses this classification system to evaluate the PRP used in each study. Considering that PRP varies in content and efficacy depending on the purification method, it is important to consider which PRP preparation is used in each study.

## 4. Basic Studies

### 4.1. Basic Studies on PRP for Intervertebral Disc Degeneration (Table 2)

Since the study examining the effects of PRP on intervertebral disc (IVD) cells was first reported in 2006 [38], several in vitro studies have been published. Many studies have used human IVD cells while others have used porcine, bovine, and rabbit cells to investigate the effects of PRP on cell growth and matrix metabolism [11]. Akeda et al. reported that PRP releasate increased the activity of the extracellular matrix metabolism of porcine nucleus pulposus and anulus fibrosus cells cultured in alginate beads [38]. Concurrently, Chen et al. concluded that growth factors in PRP, including transforming growth factor-beta1, could effectively act as a growth factor cocktail to promote the proliferation and differentiation of human nucleus pulposus cells and tissue-engineered NP formation [39]. In terms of molecular mechanisms, Kim et al. reported that PRP was effective in reducing the expression of the proteolytic enzymes matrix metalloproteinase-3 (MMP3) and cyclooxygenase-2 (COX-2), which were increased by the stimulation of inflammatory cytokines, in human intervertebral disc cells [40]. Xu et al. recently reported that PRP secreted exosomal miR-141-3p to activate the Keap1-NF-E2-related factor 2 pathway, which was found to prevent IVD degeneration [41]. In addition, PRP-derived exosomes were reported to alleviate IVD degeneration-associated inflammation by regulating the ubiquitination and autophagic degradation of the NLRP3 inflammasome [42]. Thus, exosomes have recently attracted attention in relation to PRP function, and in the study of mesenchymal stem cells (MSC) [43]. These mechanisms were summarized in Figure 1.

Several in vivo studies have been conducted in which PRP were injected into degenerated IVDs in animal models after the 2006 study by Nagae et al. reported the efficacy of PRP in IVD degeneration in a rabbit IVD model [44]. In the majority of papers, IVD degeneration models have been created in rabbits using a needle puncture to verify the effects of PRP. Obata et al. reported that PRP releasate could activate IVD cells and improve their microenvironment in rabbit IVD degeneration models [45]. Meanwhile, Chen et al. evaluated the therapeutic potential of MSC and/or PRP in miniature porcine IVD degeneration model with chymopapain [46]. By using a rat IVD degeneration model with needle puncture, Gullung et al. reported that earlier interventions of PRP in the IVD degeneration process were more beneficial than when IVD were severely degenerated [47]. In 2017, Li et al. conducted a meta-analysis of PRP animal studies and reported that IVD administration of PRP led to histological improvement in IVD degeneration and increased the magnetic resonance imaging (MRI) T2 values within IVD, which suggests that IVD degeneration was improved; the authors concluded that PRP had great potential for clinical application against IVD degeneration [48]. Although the final goal is to build up valid clinical evidence to establish PRP as an effective treatment for IVD degeneration, we should determine the molecular mechanisms of PRP in greater detail and provide patients with higher quality PRP by continuing in vitro and in vivo studies of PRP (Table 2).

### 4.2. Basic Studies of PRP in Other Spine Research Areas (Table 2)

PRP is expected to be one of the therapeutic agents capable of enhancing spinal fusion. However, the efficacy of PRP for spinal fusion remains controversial on the basis of preclinical studies [49,50,51,52,53,54]. Most previous studies have reported the results of in vivo studies, but, to the best of our knowledge, only one in vitro study has assessed the effect of PRP on osteoblasts [54]. In that study, the pharmacological activity of growth factors in freeze-dried PRP was maintained, even after four weeks of storage [54].

Kamoda, et al. reported that PRP was beneficial for both posterolateral lumbar fusion and lumbar interbody fusion in a rat model [51,53]. Meanwhile, in middle-sized animal models including rabbit, porcine, and sheep, PRP was reported to have no stimulating effect on spinal fusion [49,50,52]. Further basic studies on the effect of PRP on spinal fusions are needed in the future to reach a more accurate conclusion.

There are relatively few basic studies on PRP for treating SCI [55,56,57,58,59]. The rat SCI model was used in all studies [55,56,57,58,59]. In 2017, Salarinia et al. reported the positive effects of intrathecal PRP on nerve regeneration after SCI [55]. An additional study by Chen et al. suggested that intrathecal PRP stimulated angiogenesis, enhancing axonal regeneration after SCI in rats [56]. In 2020, Salarinia, et al. reported that a combination of PRP with MSCs synergistically promoted their therapeutic effects in the SCI [57]. Recently, Behroozi et al. concluded that human umbilical cord blood-derived PRP had the potential to reduce neuropathic pain in SCI by altering the expression of ATP receptors, and could induce motor function recovery and axonal regeneration after SCI [58,59]. However, no evidence has yet been reported in basic studies on the superiority of PRP over other treatments.

**Table 2 ijms-24-07677-t002:** Basic studies of PRP.

Research Area	Experiment Type	Author (Year)	Agent	Target	Species	Effects
IVD	In vitro	Akeda et al. (2006) [38]	Porcine PRP	IVD cells	Porcine	cell proliferation↑ collagen synthesis↑ proteoglycan accumulation↑
		Chen et al. (2006) [39]	Human PRP	NP cells	Human	cell proliferation↑ collagen synthesis↑ proteoglycan accumulation↑
		Kim et al. (2014) [40]	Human PRP	NP cells	Human	matrix metalloproteinase-3 ↓ cyclooxygenase-2 (COX-2)↓
		Xu et al. (2021) [41]	PRP-derived exosomes	NP cells IVD degeneration	Mice	exosomal miR-141-3p↑ Keap1-NF-E2-related factor 2 pathway↑
		Qian et al. (2022) [42]	Rat PRP and PRP-derived exosomes	NP cells IVD degeneration	Rat	inflammatory responses↓ IL-1β secretion↓
	In vivo	Nagae et al. (2007) [44]	Rabbit PRP	IVD degeneration/needle puncture	Rabbit	IVD degeneration↓
		Obata et al. (2012) [45]	Rabbit PRP	IVD degeneration/needle puncture	Rabbit	IVD degeneration↓
		Chen et al. (2009) [46]	Porcine MSC and/or PRP	IVD degeneration/chymopapain	Porcine	IVD degeneration↓
		Gullung et al. (2011) [47]	PRP	IVD degeneration/needle puncture	Rat	IVD degeneration↓
Spinal fusion	In vitro	Kinoshita et al. (2020) [54]	Rodent fresh or freeze-dried PRP	Osteoblast	Human	osteoblast proliferation↑
	In vivo	Kamoda et al. (2012) [51]	Rat PRP	Interbody fusion	Rat	bone union↑
		Kamoda et al. (2013) [53]	Rat PRP	Posterolateral fusion	Rat	bone union↑
		Cinotti et al. (2013) [52]	Rabbit PRP	Posterolateral fusion	Rabbit	bone union→
		Li et al. (2004) [49]	Carbon fiber cage loaded with bioceramics and platelet-rich plasma	Interbody fusion	Porcine	bone union→
		Scholz et al. (2010) [50]	Cages augmented with mineralized collagen and PRP	Interbody fusion	Sheep	bone union→
Spinal cord	In vivo	Salarinia et al. (2017) [55]	Rat PRP	Spinal cord injury	Rat	nerve regeneration↑
		Chen et al. (2018) [56]	n.d.	Spinal cord injury	Rat	locomotor recovery with neuronal regeneration
		Salarinia et al. (2020) [57]	Rat PRP and MSC	Spinal cord injury	Rat	synergistic effects in spinal cord injury
		Behroozi et al. (2021) [58]	Human umbilical cord blood-derived PRP	Spinal cord injury	Rat	neuropathic pain↓
		Behroozi et al. (2022) [59]	Human umbilical cord blood-derived PRP	Spinal cord injury	Rat	motor function recovery and axonal regeneration

PRP: platelet-rich plasma; IVD: intervertebral disc; NP: nucleus pulposus; IL-1β: interleukin-1beta; n.d.: not described; ↑: increase; →: no change; ↓: decrease.

## 5. Clinical Studies

### 5.1. Clinical Application of PRP for Intradiscal Therapy

A clinical study on PRP for intradiscal therapy was first reported in 2016 by Tuakli-Wosornu et al. [60]. Since then, 13 clinical studies or case reports have been reported (Table 3). Two randomized controlled studies, five prospective cohort studies, three retrospective cohort studies, and three case reports have been published. Most of the target diseases were discogenic LBP; however, there was one study for each targeted lumbar disc herniation (LDH) [61] and cervical degenerative disc disease [62].

Lr-PRP was used in five studies, and Lp-PRP was used in eight studies. According to the Mishra classification [37], type 1 was found in five studies, type 3 in four studies, and type 4 in three studies.

Soluble releasate isolated from activated PRP (PRP-releasate), but not PRP itself, was used in two studies [63,64]. PRP classification by Mishra et al. [37] revealed that a wide variety of PRP has been utilized for intradiscal treatments. PRP isolation kits were used in nine studies on the isolation method of PRP. PRP was manually isolated in two studies [63,64].

In all reported studies, PRP was intradiscally administrated into the targeted discs and the follow-up period varied from 3 months to 6.57 years. Tuakli-Wosornu et al. [60] conducted a prospective, double-blinded, randomized controlled study to determine the efficacy of PRP in symptomatic degenerated IVDs. Participants who received intradiscal PRP showed significantly greater improvements in functional rating index (FRI), numeric rating scale (NRS), and North American Spine Society (NASS) satisfaction scores compared to those who received a contrast agent during the eight weeks post-injection. A randomized, double-blind, active-controlled clinical trial was conducted to evaluate the efficacy and safety of an intradiscal injection of PRP-releasate compared with corticosteroid (CS) injection in discogenic LBP patients [64]. This clinical study by Akeda et al. [64] showed a clinically significant improvement in the extent of LBP evaluated using a visual analog scale (VAS) in both the PRP-releasate and CS groups at 8 weeks post-injection; however, no significant differences were found between the groups. Nevertheless, PRP-releasate injection therapy was reported to be safe and maintained improvements in LBP, disability, and QOL during the 60-week follow-up.

Four prospective cohort studies revealed that a single injection of PRP or PRP-releasate induced significant improvements in pain, disability, and quality of life (QOL) during the observational period (from 3 to 12 months) [63,65,66,67]. Among them, one study by Jian et al. [65] reported that improvements in NRS and Oswestry Disability Index (ODI) scores were positively correlated with the platelet concentration of PRP (Mishra classification: 3B). Two clinical studies evaluated the long-term effect of PRP or PRP-releasate treatment in patients with discogenic LBP. Both studies reported that the treatments had a safe and efficacious impact on improving LBP and LBP-related disability during the five to nine years of follow-up [68,69].

Recently, Jiang et al. [61] retrospectively evaluated the effect of transforaminal endoscopic lumbar discectomy (TELD) with PRP injection for patients with lumbar disc herniation. They reported that TELD with PRP treatment significantly improved LBP and LBP-related disability, MRI findings, and lowered the recurrence rate of LDH compared with the control (TELD without PRP treatment) group. Kawabata et al. [70] evaluated the safety and efficacy of PRP administration in two discogenic LBP patients with Modic type 1 change, known to be an MRI biomarker of LBP [71]. They reported that PRP injection into targeted discs with Modic type 1 change was safe and showed a tendency to alleviate LBP.

In summary, intradiscal injection therapy of PRP for degenerative disc disease is safe and shows promise for improving pain, disability, and QOL.

**Table 3 ijms-24-07677-t003:** Clinical application of PRP for intradiscal therapy.

Author (Year)	Study Design	Disease	Number of Subjects	PRP Classification	PRP Isolation Method	Outcomes	Follow Up	Results
Tuakli-Wosornu et al. (2016) [60]	RCT	DLBP	47 (29 PRP, 18 control)	1A	Kit	FRI, NRS, SF-36, modified NASS outcome questionnaire	12 months	5
Levi et al. (2016) [66]	Prospective cohort study	DLBP	22	1A	Kit	VAS, ODI	6 months	4
Akeda at al. (2017) [63]	Prospective cohort study	DLBP	14	4B	Manual	VAS, RDQ, X-ray, MRI	10 months	4
Cheng J (2019) [69]	Retrospective cohort study	DLBP	29	1A	Kit	NRS, SF-36	6.57 years	4
Wu TJ (2020) [72]	Case reports	DLBP	2	3B	Kit	VAS	3 months	3
Jain D (2020) [65]	Prospective cohort study	DLBP	25	3B	Kit	NRS, ODI	6 months	4
Akeda K at al. (2022) [64]	RCT	DLBP	15	4B	Manual	VAS, ODI, RDQ, Radiographic measurements, MRI	12 months	4
Jiang Y (2022) [61]	Prospective cohort study	LDH (TELD)	108	n.d.	Kit	VAS, ODI, MRI	12 months	5
Akeda K at al. (2022) [68]	Retrospective cohort study	DLBP	11	4B	Manual	VAS, RDQ, Radiographic measurements	5.9 years	4
Lutz C at al. (2022) [73]	Retrospective cohort study	DLBP	37	1A/3A	Kit	NRS, FRI, NASS patient satisfaction index	18 months	4
Lam et al. (2022) [62]	Case reports	CLBP	1	3-	n.d.	NRS, NDI	9 months	3
Zhang et al. (2022) [67]	Prospective cohort study	DLBP	31	3B	Kit	SF-36, FRI, NRS	48 week	4
Kawabata et al. (2023) [70]	Case reports	DLBP	2	1B	Kit	VAS, ODI, RDQ, Radiographic measurements, MRI	25 weeks	3

PRP: platelet-rich plasma; RCT: randomized controlled trial; DLBP: discogenic low back pain; LDH: disc herniation; TELD: endoscopic lumbar discectomy; CLBP: chronic low back pain; FRI: functional rating index; NRS: numeric rating scale; SF-36: MOS Short-Form 36-Item Health Survey; NASS: North American Spine Society; VAS: Visual Analogue Scale; ODI: Oswestry Disability Index; RDQ: Roland–Morris Disability Questionnaire; MRI: Magnetic Resonance Imaging; NDI: Neck Disability Index; n.d.: not described. Results 3: Trend of improvement reported, 4: Significant improvement reported, 5: Significant improvement reported compared to (placebo/sham/vehicle) control group.

### 5.2. Clinical Application of PRP for Spinal Fusion Surgery

Clinical application of PRP for spinal fusion surgery was first reported in 2003 by Weiner and Walker [74]. Since then, 17 clinical studies have been conducted (Table 4). Five randomized controlled studies, two nonrandomized studies, six prospective cohort studies, and four retrospective cohort studies have been published. The patients in 16 studies received lumbar spinal surgeries (10 posterolateral lumbar fusions [PLFs] and six interbody fusions). Only one anterior cervical discectomy and fusion was performed as a cervical spinal surgery [75].

Lr-PRP was used in 11 studies, and Lp-PRP in four studies. According to the Mishra classification [37], type 2 was found in 11 studies, and type 4 in four studies. PRP isolation kits were used in six studies. PRP was manually isolated in 11 studies. PRP was activated before surgery in all studies. The bone fusion rate was assessed in all studies using radiography and/or computed tomography (CT). The follow-up period varied from 6 to 34 months. Six studies (35.5% of total) reported that the use of PRP significantly increased the bone fusion rate compared to the control group; however, in seven studies the use of PRP showed no significant improvement in the bone fusion rate. Furthermore, two studies reported that the use of PRP in PLF surgery decreased the bone fusion rate.

Kubota et al. [76] conducted a prospective randomized controlled study with a 2-year follow-up to evaluate the efficacy of PRP after PLF surgery. Sixty-two patients who underwent one- or two-level instrumented PLF for lumbar degenerative spondylosis with instability were stratified into either the PRP (31 patients) or control (31 patients) group. PRP-treated patients underwent surgery using an autograft local bone. This clinical study showed that the bone fusion rate at the final follow-up was significantly higher in the PRP group (94%) than in the control group (74%). Moreover, they reported that the area of fusion mass was significantly higher in the PRP group than in the control group. The mean period necessary for the fusion in the PRP group was shorter than that of the control group.

Imagama et al. [77] reported on the efficacy of PRP on the rate and extent of bone fusion in PLF surgery using autologous local bone graft and PRP and the safety of PRP application during a follow-up period of 10 years. Local application of PRP combined with autologous local bone had a positive impact on early fusion in lumbar arthrodesis. They also reported that there were no adverse symptoms and events related to PRP, including seroma, and no massive bone formation or deep infection visible on MRI over the 10 year follow-up.

In contrast, two studies reported that the use of PRP in PLF surgery caused a decreased bone fusion rate. Weiner and Walker [74] reported a retrospective cohort study that evaluated the bone fusion rate in PLF surgery using autograft bone combined with PRP. The fusion rate for the control group was 91% (24 of 27). The fusion rate for the PRP group was 62% (18 of 32). They concluded that bone fusion rates using autograft bone alone were significantly higher than those using autograft combined with PRP (*p* < 0.05).

Acebal-Cortina [78] conducted a prospective controlled blinded non-randomized study to analyze if adding the PRP to a mixture of local autograft plus tricalcium phosphate and hydroxyapatite (TCP/HA) would improve the fusion rate in PLF surgery. They reported that correct fusion was seen in 93% of the cases (37 of 40) in the control group. In the PRP group, correct fusion was seen in 75% of the cases (50 of 67). They concluded that the addition of PRP to a mixture of autologous bone graft plus TCP/HA decreased the fusion rate of PLF.

Sys et al. [79] conducted a prospective randomized controlled study to assess the radiological effect of PRP when added to autograft iliac crest bone in mono-segmental posterior lumbar interbody fusion. PRP was produced using an isolation kit and activated with thrombin (1000 U/mL in 10% CaCl_2_). Then, the cages were filled with autologous bone chips and steeped in a plasma-thrombin solution until clotting visually occurred (approximately 10 min). However, the authors concluded that adding PRP in posterior lumbar interbody fusion did not lead to a substantial improvement or deterioration when compared to autologous bone alone.

The assessment of clinical outcomes, such as visual analog scale (VAS) of LBP, VAS of leg pain, VAS of leg numbness, the ODI, and the Short-Form 36, was performed in eight studies [75,76,79,80,81,82,83,84]. All studies reported that there was no clear benefit in terms of clinical outcomes when PRP was used in spinal surgery.

In summary, the effectiveness of PRP in spinal fusion surgery is limited. Whether the addition of PRP to autologous bone grafts increases the bone fusion rate remains controversial, and there were no differences in the clinical outcomes between PRP and control groups.

**Table 4 ijms-24-07677-t004:** Clinical application of PRP for spinal fusion surgery.

Author (Year)	Study Design	Surgical Procedure	Number of Subjects (PRP/Control)	PRP Classification	PRP Isolation Method	Outcomes	Follow Up	Results
Weiner and Walker (2003) [74]	Retrospective cohort study	PLF	32/27	2A	Kit	Bone fusion rate	24 months	1
Hee et al. (2003) [80]	Prospective cohort study	TLIF	23/111	2B	Manual	Bone fusion rate	25 months [24–27]	2
Castro et al. (2004) [85]	Prospective cohort study	TLIF	22/62	2B	Manual	Bone fusion rate	34 ± 2 months	3
Carreon et al. (2005) [86]	Retrospective cohort study	PLF	76/76	2-	Manual	Bone fusion rate	32 months [24–48]	2
Jenis et al. (2006) [81]	Prospective cohort study	anterior spinal fusion (interbody)	15/22	2-	Manual	Bone fusion rate	25.7 months [6–40]	2
Feiz-Erfan et al. (2007) [75]	RCT	ACDF	42/39	2A	Kit	Bone fusion rate	24 months	2
Tsai et al. (2009) [82]	RCT	PLF	34/33	n.d.	Manual	Bone fusion rate	28.5 months [24–34.9]	2
Hartmann et al. (2010) [83]	Retrospective cohort study	anterior spinal fusion (interbody)	15/20	2-	Kit	Bone fusion rate, bone density	8.3 months [4–15]	5
Acebal-Cortina et al. (2011) [78]	Nonrandomized study	PLF	67/40	n.d.	Manual	Bone fusion rate	24 months	1
Landi et al. (2011) [87]	Prospective cohort study	PLF	14/14	2A	Kit	Bone fusion rate, bone density	6 months	3
Sys el al. (2011) [79]	RCT	PLIF	19/19	2A	Kit	Bone fusion rate	12 months	2
Tarantino et al. (2014) [88]	Prospective cohort study	PLF	20/20	2-	Kit	Bone fusion rate, bone density	12 months	5
Vadalà et al. (2016) [89]	Nonrandomized study	PLF	10/10	2-	Manual	Bone fusion rate	12 months	5
Rezende et al. (2017) [90]	RCT	PLF	20/20	4-	Manual	Bone fusion rate	6 months	2
Imagama et al. (2017) [77]	Prospective cohort study	PLF	29/29	4A	Manual	Bone fusion area	12 months	5
Kubota et al. (2018) [84]	Retrospective cohort study	TLIF	11/9	4A	Manual	Duration of bone fusion, and bone fusion rate	24 months	5
Kubota et al. (2019) [76]	RCT	PLF	25/25	4A	Manual	Duration of bone fusion, bone fusion rate, and bone fusion area	24 months	5

PRP: platelet-rich plasma; RCT: randomized controlled trial; PLF: posterolateral lumbar fusion; TLIF: transforaminal lumbar interbody fusion; ACDF: anterior cervical discectomy and fusion; PLIF: posterior lumbar interbody fusion; n.d.: not described. Results 1: Worsening reported, 2: No clear benefit reported, 3: Trend of improvement reported, 5: Significant improvement reported compared to (placebo/sham/vehicle) control group.

### 5.3. Clinical Application of PRP for Intraarticular Therapy of Facet or Sacroiliac Joint Pain

A clinical study of PRP for the treatment of lumbar facet joint syndrome was first reported in 2016 by Wu et al. [91]. Since then, three clinical studies or case reports have been reported (Table 5). One randomized controlled study and two case series have been published. Most of the targeted diseases were regarding lumbar facet joint pain; however, in one study addressing chronic LBP, multiple site injection was performed.

Lr-PRP was used in two studies, and Lp-PRP in one study. According to the Mishra classification [37], type 1 was found in two studies and type 4 in one study. PRP isolation kits were used in one study, and manual isolation in two studies [91,92].

**Table 5 ijms-24-07677-t005:** Clinical application of PRP for intraarticular therapy for facet joint pain.

Author (Year)	Study Design	Disease	Number of Subjects	PRP Classification	PRP Isolation Method	Outcomes	Follow Up	Results
Wu et al. (2016) [91]	Case series	Facet joint syndrome	19	1B	Manual	Pain VAS, RDQ, ODI, Modified MacNab criteria	3 months	4
Kirchner et al. (2016) [93]	Case series	LBP	86	4B	Kit	VAS	6 months	4
Wu et al. (2017) [92]	RCT	Facet joint syndrome	46	1B	Manual	Pain VAS, RDQ, ODI, Modified MacNab criteria	6 months	5

PRP: platelet-rich plasma; RCT: randomized controlled trial; VAS: Visual Analogue Scale; RDQ: Roland–Morris Disability Questionnaire; ODI: Oswestry Disability Index. Results 4: Significant improvement reported, 5: Significant improvement reported compared to (placebo/sham/vehicle) control group.

PRP was administrated into the facet joint under x-ray fluoroscopic control in all the reported studies. The follow-up period varied from three to six months. Wu et al. [92] conducted a prospective, double-blinded, randomized controlled study to determine the efficacy of PRP in lumbar facet joint syndrome. Both PRP injection and local anesthetic (LA)/corticosteroid (CS) injection were determined to be effective, easy, and safe enough for the treatment of lumbar facet joint syndrome after six months of follow-up. However, autologous PRP had better outcomes than LA/CS for the duration of treatment efficacy.

One case series [93] evaluated the multiple site injections (intradiscal, facet joint and/or epidural space) of PRP for the treatment of chronic LBP and reported significant improvement after injection.

A clinical study of PRP for the treatment of sacroiliac joint pain was first reported in 2016 by Navani et al. [94]. Since then, seven clinical studies or case reports have been reported (Table 6). Two randomized controlled studies, one non-randomized controlled study, and four case series have been published.

Lr-PRP was used in five studies, and Lp-PRP was used in one study. According to the Mishra classification [37], type 1 was found in four studies, type 2 in one study, and type 3 in one study. PRP isolation kits were used in five studies on the isolation methods of PRP; PRP was manually isolated in two studies [94,95].

PRP was reportedly administered into the sacroiliac joint under ultrasound guidance in four studies, and under fluoroscopic guidance in three studies. The follow-up period varied from three months to four years. Singla et al. [95] conducted a prospective, randomized, open-label, blinded-endpoint (PROBE) study to determine the efficacy of PRP in 40 patients with sacroiliac joint pain. The reduction in pain intensity and improvements in functional disability were significantly greater and lasted longer in the PRP group compared to the steroid group. In contrast, Chen et al. [96] conducted a prospective randomized double-blinded clinical trial in 26 patients with sacroiliac joint pain and with a positive diagnostic block. The results of the study showed that both PRP and corticosteroid groups showed improvements in pain and function; however, the steroid group had a significantly greater response and more responders than the PRP group. Eldin et al. [97] conducted a non-randomized controlled trial to compare platelet concentrates (PRP and platelet-rich fibrin [PRF]) in injectable form in sacroiliac joint dysfunction. This study showed a clinically significant improvement in the extent of LBP evaluated by a visual analog scale (VAS) in both the PRP and PRF groups; however, the reduction in pain intensity lasted longer in the PRF group than in the PRP group. Four case series revealed that a single injection of PRP induced significant improvements in pain, disability, or QOL during the observational period (from 6 to 48 months) [94,95,98,99].

In summary, an injection therapy of PRP for the patients with facet joint or sacroiliac joint pain is safe and useful for improving pain, disability, and QOL.

**Table 6 ijms-24-07677-t006:** Clinical application of PRP for intraarticular therapy for the sacroiliac joint pain.

Author (Year)	Study Design	Disease	Number of Subjects	PRP Classification	PRP Isolation Method	Outcomes	Follow Up	Results
Navani et al. (2016) [94]	Case series	chronic SIJ pain	10	n.d.	Manual	VAS, SF-36	12 months	3
Singla et al. (2017) [95]	RCT	chronic low back pain	40	3	Manual	VAS, MODQ, SF-12	3 months	5
Ko et al. (2017) [99]	Case reports	SIJ pain	4	1A	Kit	SFM, NRS, and ODI	48 months	4
Eldin et al. (2019) [97]	Non-RCT	SIJ dysfunction pain	PRF 124 PRP 62	PRF 1 PRP 2	Kit	VAS	6 months	4
Wallace et al. (2020) [100]	Case series	SIJ dysfunction pain	50	1A	Kit	ODI, NRS	6 months	4
Broadhead et al. (2020) [98]	Case reports	SIJ dysfunction pain	1	1A	Kit	NRS, ODI	12 months	4
Chen et al. (2022) [96]	RCT	SIJ pain	26	1-	Kit	NRS, ODI	6 months	4

PRP: platelet-rich plasma; RCT: randomized controlled trial; SIJ: sacroiliac joint; PRF: platelet-rich fibrin; VAS: Visual Analogue Scale; SF-36: MOS Short-Form 36-Item Health Survey; MODQ: Modified Oswestry Disability Questionnaire; SF-12: MOS Short-Form 12-Item Health Survey; SFM: Short-form McGill Pain Questionnaire; NRS: numeric rating scale; ODI: Oswestry Disability Index; n.d.: not described. Results 3: Trend of improvement reported, 4: Significant improvement reported, 5: Significant improvement reported compared to (placebo/sham/vehicle) control group.

### 5.4. Clinical Application of PRP for Epidural Therapy

Regarding epidural injection therapy for spinal symptoms, eleven studies, including two randomized controlled trial (RCT) studies, four prospective cohort studies, three retrospective cohort studies, and two case series, have been reported (Table 7). Targeted symptoms were LBP and radicular pain; however, one study targeted cervical pain [101]. From the viewpoint of an epidural injection approach, a transforaminal approach was used in five studies, an interlaminar approach in two studies, both a transforaminal and interlaminar approach in one study, and a caudal (sacral hiatus) approach in three studies. PRP was injected into the epidural space with or without additional sites of PRP injection, including intradiscal, intraarticular, or intraosseous injection. The follow-up period varied from 3 to 35.7 months.

Lr-PRP was used in two studies, and Lp-PRP in five. There were four studies without detailed descriptions of PRP characteristics. According to the Mishra classification system [37], type 1 was found in two studies, type 3 in two studies, and type 4 in three studies; there was no classification assigned in the remaining studies. PRP was isolated using a commercially available kit in three studies and manually in eight studies.

Ruiz-Lopez R, et al. [102] conducted a randomized controlled double-blinded study comparing Lr-PRP and corticosteroid administered via a caudal epidural injection for chronic low back pain (LBP). The patients whose LBP with or without radiculopathy lasted for at least three months were randomly assigned to receive an epidural injection of Lr-PRP (*n* = 25) or corticosteroid (*n* = 25) into the S3-4 epidural space under fluoroscopic guidance. At one month after the epidural injection, both the corticosteroid and Lr-PRP groups showed a significant reduction in VAS; however, the Lr-PRP group showed sustained improvements at six months after treatment, while VAS in the corticosteroid groups was re-increased and reached a baseline level at six months after treatment. Furthermore, all domains of the Short Form 36-Item Health Survey (SF-36) after treatment in the Lr-PRP group were significantly higher than those in the corticosteroid group. The authors concluded that both autologous Lr-PRP and corticosteroids for caudal epidural injections are equally safe and therapeutically effective in patients with chronic LBP, and that Lr-PRP is superior to corticosteroids for achieving an increased duration of the analgesic effect and improved quality of life.

Xu Z, et al. [103] conducted an RCT to compare the efficacy and safety of transforaminal injections of PRP (*n* = 61) and steroid (*n* = 63) in patients who suffer from LBP with unilateral radicular pain due to lumbar disc herniation. Significant improvements in VAS, ODI, and other parameters were observed in both groups after one month, and were maintained for one year. There were no significant differences in all assessments between the steroid and PRP treatment groups. Bise et al. [104] conducted a prospective cohort study to compare the short-term (6 weeks) therapeutic effect of PRP versus corticosteroid by an interlaminar approach in patients with prolonged unilateral radicular pain. Patients underwent prednisolone injection (*n* = 30) or Lr-PRP injection (*n* = 30). At six weeks post-injection, both treatments equally and significantly decreased the numerical rating scale and ODI without any major complications.

PRP injection into multiple sites for patients with chronic LBP has been reported [93,101,105,106]. Kirchner et al. [93] retrospectively reported that intradiscal, intra-articular facet, and transforaminal epidural injection of PRP under fluoroscopic guidance-control significantly decreased VAS in 86 patients with chronic LBP for 6 months. They also showed that minimal clinically important differences for NRS and ODI were achieved in 47 patients with chronic LBP after intradiscal, epidural, and intraosseous PRP injection [101].

In summary, the epidural injection of PRP showed safety and efficacy for the treatment of LBP and radiculopathy. The analgesic effect induced by PRP on LBP was slower but lasted longer compared to corticosteroid injections.

**Table 7 ijms-24-07677-t007:** Clinical application of PRP for epidural therapy.

Author (Year)	Study Design	Disease	Number of Subjects	PRP Classification	PRP Isolation Method	Outcomes	Follow Up	Results
Kirchner et al. (2016) [93]	Retrospective cohort study	LBP w/or w/o radicular pain	86	4B	Kit	VAS for LBP	6 months	4
Bhatia et al. (2016) [107]	Case series	LBP w/or w/o radicular pain	10	n.d.	Manual	VAS, ODI, SLRT	3 months	3
Rawson et al. (2020) [108]	Case series	radicular pain	2	1A	Manual	Pain	3–6 months	3
Bise et al. (2020) [104]	Prospective cohort study	radicular pain	60 (Steroid 30-PRP30)	3B	Manual	NRS for leg pain, ODI	6 weeks	4
Ruiz-Lopez et al. (2020) [102]	RCT	LBP w/or w/o radicular pain	50 (Steroid 25-PRP25)	1-	Manual	VAS for LBP, SF-36	6 months	5
Machado et al. (2021) [106]	Prospective cohort study	LBP w/or w/o radicular pain	46	3B	Manual	VAS for LBP, RDQ, NASS Satisfaction	52 weeks	4
Kirchner et al. (2021) [101]	Retrospective cohort study	Cervical and low back pain	65 (18 Cervical and 47 LBP)	4B	Kit	NRS for neck and LBP, COMI, ODI	5 months [1–24]	4
Xu et al. (2021) [103]	RCT	LBP w/radicular pain	124 (Steroid 68-PRP64)	n.d.	Manual	VAS, ODI, SF-36 and etc	12 months	4
Barbieri et al. (2022) [105]	Prospective cohort study	LBP w/or w/o radicular pain	30	4-	Kit	VAS for LBP and leg pain, ODI, PGIC	6 months	2
Yalçın Demirci et al. (2022) [109]	Retrospective cohort study	Radicular pain	62 (Steroid 31-PRP31)	n.d.	Manual	VAS, ODI	35.7 months	4
Le et al. (2023) [110]	Prospective cohort study	LBP w/radicular pain	25	n.d.	Manual	VAS, ODI, SLRT	12 months	4

PRP: platelet-rich plasma; RCT: randomized controlled trial; LBP: low back pain; VAS: Visual Analogue Scale; ODI: Oswestry Disability Index; SLRT: Straight Leg Raising Test; NRS: numeric rating scale; SF-36: MOS Short-Form 36-Item Health Survey; RDQ: Roland–Morris Disability Questionnaire; NASS: North American Spine Society; COMI: Core Outcome Measure Index; PGIC: Patient Global Impression of Change; n.d.: not described. Results 2: No clear benefit reported, 3: Trend of improvement reported, 4: Significant improvement reported, 5: Significant improvement reported compared to (placebo/sham/vehicle) control group.

### 5.5. Clinical Application of PRP for Spinal Cord Injury

Several in vitro and in vivo studies showed the regenerative effects of PRP on SCI; however, only one clinical case series reported the efficacy of the administration of PRP and bone marrow aspirate concentrate (BMAC) in SCI patients [111]. Shehadi et al. conducted intrathecal and intravenous co-administration of PRP and BMAC in seven patients (age range: 22–65 years), with SCI as the salvage therapy. Injury levels ranged from C3 through T11, and the elapsed time between the injury and salvage therapy ranged from 2.4 months to 6.2 years. They reported improvements in ODI in several patients and concluded that intrathecal/intravenous co-administration of PRP and BMAC resulted in no significant complications and may have had some clinical benefits.

## 6. Future Perspectives

There are still open questions regarding the mechanism of action of PRP. For example, in future studies, key bioactive molecules that exert biological effects should be identified among the functional components included in PRP to understand the molecular mechanisms of tissue repair. This would increase the reliability of PRP in clinical use. The efficacy of PRP on other pathologies of spinal diseases, including lumbar canal stenosis, postoperative pain due to surgical tissue damage, or cervical spine diseases, should also be verified in future clinical applications. The application of allogenic and/or stem cell-derived platelets [112,113] for PRP should be considered to obtain the equivalence of therapeutic effects of PRP and to promote its commercialization in the future.

## 7. Conclusions

In this paper, based on previously published basic and clinical studies, we reviewed the effects of PRP on pathological spinal conditions, including degenerative disc disease, spinal fusion, spinal cord injury, LBP, and radicular pain. Because our primary aim was to provide a comprehensive review of the current literature on the basic mechanisms and emerging clinical applications for the treatment of several spinal diseases with PRP, we did not perform individual meta-analyses of the efficacy of PRP for treating these pathological spinal conditions. Among basic studies, it is clearly suggested that PRP is effective for treating degenerative disc disease. However, we cannot draw conclusions about the effect of PRP for spinal fusion and spinal cord injury because different studies have reported opposing results, and the number of studies is insufficient. In the future, to enhance the clinical efficacy of PRP for degenerative disc disease, more detailed basic studies are needed to further clarify the molecular mechanisms of PRP. Meanwhile, for spinal fusion and spinal cord injuries, higher quality basic studies are required to determine the effect of PRP. In clinical studies, PRP has the advantage of being safe and easily applied in a clinical setting since PRP is derived from autologous blood; however, because of individual differences in the concentration and function of platelets, it is difficult to standardize the treatment using PRP. In addition, it is even more difficult to determine the effect of PRP because it lacks uniform characteristics due to the variety of purification methods used. Therefore, we assessed the PRP used in each clinical study by Mishra’s classification to determine the effect of PRP more accurately. In this review, intradiscal injection therapy of PRP for degenerative disc disease is considered safe and effective. In contrast, the effect of PRP for spinal fusion surgery may be limited. For facet joint or sacroiliac joint pain, an injection therapy of PRP may be safe and useful; although, patient selection was a challenge in certain studies. In addition, the epidural injection of PRP also showed safety and efficacy for LBP and radiculopathy, but future studies need to include additional eligible patients and limited injection sites.

Taken together, PRP has the potential to be a breakthrough treatment for several spinal diseases. However, to establish PRP therapy as an evidence-based treatment, large-scale double-blind randomized trials with appropriate patient selection and homogeneity of PRP components are required in the future.

## Figures and Tables

**Figure 1 ijms-24-07677-f001:**
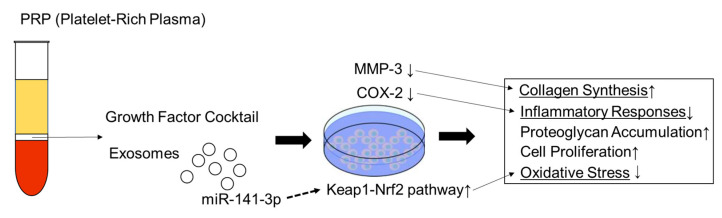
Schematic model for mechanism of PRP on intervertebral disc cells. ↑: increase; ↓: decrease.

**Table 1 ijms-24-07677-t001:** Bioactive proteins released from α-granule.

Factor	Examples
Adhesive proteins	Von Willebrand factor, fibrinogen, fibronectin, vitronectin, thrombospondin-1 and -2, laminin-8
Clotting factors and inhibitors	Factor V/Va, factor XI, multimerin, protein S, high-molecular-weight kininogen, protease nexin-1 and -2, tissue factor pathway inhibitor, protein C inhibitor
Fibrinolytic factors and inhibitors	Plasminogen, plasminogen activator inhibitor-1, urokinase-type plasminogen activator (u-PA), α2-antiplasmin, histidine-rich glycoprotein, thrombin-activatable fibrinolysis inhibitor (TAFI,) α2-macroglobulin
Proteases and antiproteases	Metalloproteinases (MMP)-1, -2, -4, -9, a disintegrin and metalloproteinase with thrombospondin motifs (ADAMTS) 10, -13, TIMPs 1–4, platelet inhibitor of FIX, C1 inhibitor, α1-antitrypsin
Growth and mitogenic factors	transforming Growth Factor (TGF)-β1, -β2, platelet-derived growth factor (PDGF) -A, -B, and -C, epithelial growth factor (EGF), insulin-like growth factor-1 (IGF-1), vascular endothelial growth factor (VEGF) -A, -C, basic fibroblast growth factor (bFGF)-2, hepatocyte growth factor (HGF), bone morphometric protein (BMP)-2, -4, -6, connective tissue growth factor (CTGF), signal peptide, CUB domain and EGF-like domain containing 1 (SCUBE1), insulin-like growth factor binding protein 3 (IGFBP3)
Chemokines, cytokines and others	Interleukin (IL)-1, RANTES (CCL5), IL-8 (CXCL8), macrophage inflammatory protein (MIP)-1α (CCL3), MIP-2 (CXCL2), LIX (CXCL6) GRO-α (CXCL1), ENA-78 (CXCL5), stromal cell-derived factor (SDF)-1α (CXCL12), MCP-1 (CCL2), MCP-3 (CCL7), platelet factor 4 (PF4) (CXCL4), pro-platelet basic protein (PBP), β-thromboglobulin (β-TG), neutrophil activating protein-2 (NAP-2), connective-tissue activating peptide III T(CXCL7), thymus and activation-regulated chemokine (TARC) (CCL17), angiopoietin-1, high mobility group box 1 (HMGB1), interleukin-6 soluble receptor (IL-6sR), bone sialoprotein, dickkopf-1, osteoprotegerin
Others	Chondroitin 4-sulfate, albumin, immunoglobulins G and M, amyloid β-protein precursor, disabled-2, complement factor H, bile salt-dependent lipase (BSDL), semaphorin 3A

This table was cited from [16] with modifications (Reprinted with permission from Georg Thieme Verlag KG).

## Data Availability

Not applicable.

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
