# Peer review of "Advances in Platelet-Rich Plasma Treatment for Spinal Diseases: A Systematic Review"

_ijms, 2023, doi:10.3390/ijms24087677_

Round 1
Reviewer 1 Report
Suggestion: Minor revisions
The authors have put together a well-researched review for the use of PRP in spinal injuries and applications, all in one place. The aim of the review is very clear from the beginning and is informative. Here are some minor suggestions to further strengthen the paper.
Minor suggestions:
1. Please add the following 3 references to further strengthen your introduction, specifically line 51 and 52- ‘Biological and/or cellular therapies have been utilized in a variety of regenerative medicine treatments. Platelet-rich plasma (PRP) has been used clinically for tissue regeneration and repair’:
a. 2016, Platelet-rich plasma, a source of autologous growth factors and biomimetic scaffold for peripheral nerve regeneration 10.1080/14712598.2017.1259409
b. 2018, Stem-cell therapy and platelet-rich plasma in regenerative medicines: A review on pros and cons of the technologies 10.4103/jomfp.JOMFP_93_18
c. 2022, Effect of Combined Intraosseous and Intraarticular Infiltrations of Autologous Platelet-Rich Plasma on Subchondral Bone Marrow Mesenchymal Stromal Cells from Patients with Hip Osteoarthritis https://doi.org/10.3390/jcm11133891
2. Please re-format Tables 6 and 7 in alignment with the rest of the tables.

Minor editing of English language required
Author Response
The authors have put together a well-researched review for the use of PRP in spinal injuries and applications, all in one place. The aim of the review is very clear from the beginning and is informative. Here are some minor suggestions to further strengthen the paper.
Response) We wish to extend our gratitude to the reviewer for taking time to review our manuscript and giving us comments.
Minor suggestions:
- Please add the following 3 references to further strengthen your introduction, specifically line 51 and 52- ‘Biological and/or cellular therapies have been utilized in a variety of regenerative medicine treatments. Platelet-rich plasma (PRP) has been used clinically for tissue regeneration and repair’:
- 2016, Platelet-rich plasma, a source of autologous growth factors and biomimetic scaffold for peripheral nerve regeneration 10.1080/14712598.2017.1259409
- 2018, Stem-cell therapy and platelet-rich plasma in regenerative medicines: A review on pros and cons of the technologies 10.4103/jomfp.JOMFP_93_18
- 2022, Effect of Combined Intraosseous and Intraarticular Infiltrations of Autologous Platelet-Rich Plasma on Subchondral Bone Marrow Mesenchymal Stromal Cells from Patients with Hip Osteoarthritis https://doi.org/10.3390/jcm11133891
Response) According to the reviewer’s comment, three papers were cited at the specified lines in the Introduction section (highlighted in gray).
- Please re-format Tables 6 and 7 in alignment with the rest of the tables.
Response) It was corrected accordingly.
Reviewer 2 Report
1. Line 32: The authors mentioned that "Among spinal disorders, low back pain (LBP) is one of the most common musculoskeletal conditions", but I'm afraid that low back pain is just a description of symptoms, which consists of a variety of spinal diseases. Many patients with LBP lack a definite diagnosis, and thus not able to be treated with PRP.
2. Line 128-129: Why does PRP need to be activated to treat hard tissue but not the case in the soft tissue? The way to activate PRP should also be mentioned in the manuscript.
3. Line 142-144: P-PRP appeared twice and L-PRP appeared three times for different abbreviated targets. How come different kinds of PRP have the same abbreviation?
4. Line 155-173: The whole story of the in vitro mechanism behind the action of PRP on cells of intervertebral disc deserves a well-drawn picture to illustrate it intead of just a lengthy paragraph and table.
5. Line 229: What is PRP releasate and how does it prepared? The authors should provide detailed explanation on it in the paragraph.
6. Line 251: Amon is a typo.
7. Line 267: There are already two randomized controlled studies in your review. Why did the authors conclude here that further randomized controlled studies are still needed in the future? Were there still any issues not been fully addressed by the two randomized controlled studies?
8. Line 394: Why did the authors concluded here that further RCTs with larger sample size is needed? RCTs done by Wu et al and Singla et al already revealed significant improvement comparing to control group. Didn't they have enough subjects during sample size calculation before study recruitment?
9. Line 424: According to the Mishra classification, there are type 1-4 instead of A and B. The authors should categorize PRP used in these studies into type 1-4 in the manuscript.
10. Line 461: The authors stated that further accumulation of evidence is need. However, what about the odds ratio and the confidence interval of the pooled effect of current studies? How does the authors know that the current evidence is still insufficient?
Author Response
We wish to extend our gratitude to the reviewers for taking the time to review our manuscript and providing valuable suggestions that have helped us improve the quality of our paper.
- Line 32: The authors mentioned that "Among spinal disorders, low back pain (LBP) is one of the most common musculoskeletal conditions", but I'm afraid that low back pain is just a description of symptoms, which consists of a variety of spinal diseases. Many patients with LBP lack a definite diagnosis, and thus not able to be treated with PRP.
Resposne) Thank you very much for pointing out an important issue. The authors agree with the reviewer’s notion. Therefore, the description was revised as follows (highlighted in gray).
Page 1, line 32:
Low back pain (LBP) is one of the most common complaints of patients with spinal diseases.
- Line 128-129: Why does PRP need to be activated to treat hard tissue but not the case in the soft tissue? The way to activate PRP should also be mentioned in the manuscript.
Response) Thank you for the reviewer’s valuable comments. Activated PRP has also been injected into the soft tissue. Therefore, the description regarding this was deleted. The method of PRP activation was described as follows (highlighted in gray).
Page 4, lines 136 to 138:
Exogenous activation techniques of PRP have been used for in vivo and clinical studies. PRP is usually activated by the addition of calcium chloride and/or thrombin, freezing and thawing, or exposure to collagen [11].
- Line 142-144: P-PRP appeared twice and L-PRP appeared three times for different abbreviated targets. How come different kinds of PRP have the same abbreviation?
Response) The whole manuscript defined Leukocyte-poor PRP as Lp-PRP and Leukocyte-rich PRP as Lr-PRP.
- Line 155-173: The whole story of the in vitro mechanism behind the action of PRP on cells of intervertebral disc deserves a well-drawn picture to illustrate it intead of just a lengthy paragraph and table.
Response) According to the reviewer’s comment, the illustration (Figure 1) was added to section 4.1.
- Line 229: What is PRP releasate and how does it prepared? The authors should provide detailed explanation on it in the paragraph.
Response) PRP releasate was isolated from activated PRP followed by centrifugation. The description was revised as follows.
Page 8, lines 243 to 244:
Soluble releasate isolated from activated PRP (PRP-releasate), but not PRP itself, was used in two studies [60, 61].
- Line 251: Amon is a typo.
Response) It was corrected as ‘Among’.
- Line 267: There are already two randomized controlled studies in your review. Why did the authors conclude here that further randomized controlled studies are still needed in the future? Were there still any issues not been fully addressed by the two randomized controlled studies?
Response) Thank you very much for the reviewer’s thoughtful suggestion. The authors agree with the reviewer’s notion. Therefore, the following description in section 5.1. was deleted.
; however, future randomized controlled clinical studies should be performed to evaluate the clinical effects of this therapy systematically
- Line 394: Why did the authors concluded here that further RCTs with larger sample size is needed? RCTs done by Wu et al and Singla et al already revealed significant improvement comparing to control group. Didn't they have enough subjects during sample size calculation before study recruitment?
Response) The authors agree with the reviewer’s notion. Therefore, the following description in section 5.3. was deleted.
; however, future randomized controlled clinical studies in large sample size should be required to establish the optimal method of PRP preparation and selection of appropriate patients
- Line 424: According to the Mishra classification, there are type 1-4 instead of A and B. The authors should categorize PRP used in these studies into type 1-4 in the manuscript.
Response) Mishra classification with types 1 to 4 was added in sections 5.1. to 5.4. (highlighted in gray)
- Line 461: The authors stated that further accumulation of evidence is need. However, what about the odds ratio and the confidence interval of the pooled effect of current studies? How does the authors know that the current evidence is still insufficient?
Response) Thank you very much for the reviewer’s comments. The authors decided to delete the following description in section 5.4.
However, further accumulation of evidence including randomized controlled studies with eligible patient selection and limited injection site is required to establish the efficacy of PRP epidural injection therapy for LBP and radiculopathy.
Reviewer 3 Report
In this review the authors have performed a careful analysis of the basic and clinical studies exploiting platelet-rich plasma (PRP) as a therapeutic tool for various backbone diseases causing pain and dishability. Thje topic is original and appropriate for the IJMS special issue it is dedicated to. I have listed below the points which require the authors’ attention in view of resubmission of an amended version.
Major points
1) Section 3: a description of the currently employed methods to obtain PRP upon blood sampling should be added, especially for the open method, namely: anticoagulants, preferred type of tubes, minimal blood sample volume, procedure to separate blood cells, centrifugation parameters (g, min.), etc.
2) Section 3: additional information on the purportedly optimal concentrations in absolute values (e.g. as No./microliter) of platelets in PRP and of leukocytes in Lr-PRP should be given here. Perhaps a related information is reported in Table 2, but the meaning of the two columns, A and B, under ‘Platelet concentration’ is obscure (I guess that 5x could mean 5-fold more concentrated than in normal blood, but this is not an absolute measure). After reading this section, a general reader should be able to understand the amount - as a single value or range of values - of platelets in PRP to achieve positive effects when used in clinics. This is particularly important because different methods are used for PRP production.
3) Section 4: a brief description of the methods to induce IVD injury (mechanical? enzymatic?) in experimental animals should be added here. This may help to understand which kind of reaction to injury can be expected.
4) Section 4, line 186: please explain that T2 signal in MRI is closely related to water content (e.g. inflammatory oedema if higher or nucleus polposus sclerosis if lower).
5) Section 5.1, line 240: please explain better why the PRP-treated arm of patients was compared with a contrast agent-treated arm.
6) Section 5.1: if available in the analyzed studies, objective comparison between PRP-treated patients and untreated controls by NMR or CT-scan should be reported and discussed.
Minor points
7) Page 2, section 2.1, line 69: please replace ’secretory organelle’ with ‘secretory inclusions’: usually, organelles are intended as the subcelluar structures needed for cell life.
8) Page 2, section 2.1, line 82, please rephrase as: ‘Platelets contain 3-8 dense granules, which contain high concentrations…’ indeed, dense granules are likely formed by the RER-Golgi secretory pathway and not by the endosomal-lysosomal pathway
9) Page 4, line 123, please spell out ‘PAW’; line 136, PPR should be PRP
10) Page 5, line 207: please replace ‘neuronal regeneration’ with ‘axonal regeneration’. It is not believable that spinal cord or ganglionic neurons may actually regenerate upon PRP stimulation since they are a well-known post-mitotic cell population.
Author Response
In this review the authors have performed a careful analysis of the basic and clinical studies exploiting platelet-rich plasma (PRP) as a therapeutic tool for various backbone diseases causing pain and dishability. Thje topic is original and appropriate for the IJMS special issue it is dedicated to. I have listed below the points which require the authors’ attention in view of resubmission of an amended version.
Response) We wish to extend our gratitude to the reviewers for taking the time to review our manuscript and providing valuable suggestions that have helped us improve the quality of our paper.
Major points
1) Section 3: a description of the currently employed methods to obtain PRP upon blood sampling should be added, especially for the open method, namely: anticoagulants, preferred type of tubes, minimal blood sample volume, procedure to separate blood cells, centrifugation parameters (g, min.), etc.
Response) We agree with the reviewer’s suggestion. Various PRP purification methods have been reported; therefore, we added the following sentence regarding the differences in PRP purification methods. The authors appreciate the reviewer’s understanding.
Page 4, lines 117 to 127:
Kushida et al. [23] compared seven systems and evaluated the purified PRP in detail. Centrifugation was performed two times in four of the systems and once in three of the systems, each following the original protocol for PRP preparation. PRP was separated by tube centrifugation in four systems, gel separation in two systems, and fully automated centrifugation in one system. The required whole blood volume ranged from 8 to 60 mL, the final volume of PRP ranged from 0.6 to 3 mL, and the average platelet concentration of PRP varied widely from 8.8 x 104/µL to 152.1 x 104/µL, depending on the system. The mean platelet-derived growth factor-AB (PDGF-AB) concentration of PRP was also reported to vary depending on the system. Platelet concentration ratios from less than 2-fold to an 8.5-fold increase have been reported over baseline; however, reports suggest a 3- to 5-fold increase is desirable [23, 24].
2) Section 3: additional information on the purportedly optimal concentrations in absolute values (e.g. as No./microliter) of platelets in PRP and of leukocytes in Lr-PRP should be given here. Perhaps a related information is reported in Table 2, but the meaning of the two columns, A and B, under ‘Platelet concentration’ is obscure (I guess that 5x could mean 5-fold more concentrated than in normal blood, but this is not an absolute measure). After reading this section, a general reader should be able to understand the amount - as a single value or range of values - of platelets in PRP to achieve positive effects when used in clinics. This is particularly important because different methods are used for PRP production.
Response) We appreciate the reviewer’s helpful comment. Because the platelet concentration of PRP depends on the platelet concentration of whole blood, we used Mishra's classification, which uses a ratio to whole blood. Although effective platelet counts have been reported to be 3-5 times higher than in whole blood, no consensus has yet been reached. Also, there is no consensus on white blood cell counts in PRP, as noted in lines 140 to 145. One of the purposes of this review is to evaluate whether differences in platelet concentrations and white blood cell counts make a difference in the effectiveness of PRP. In addition, the following description was added to the Table 2 legend (highlighted in gray).
Table 2:
Platelet concentration: subtype A contains an increased platelet concentration at or above five times baseline. Subtype B contains an increased platelet concentration less than five times baseline.
3) Section 4: a brief description of the methods to induce IVD injury (mechanical? enzymatic?) in experimental animals should be added here. This may help to understand which kind of reaction to injury can be expected.
Response) We agree with the reviewer’s suggestion. We have added the description of the methods to induce IVD degeneration in experimental animals in Table 3 (highlighted in gray).
4) Section 4, line 186: please explain that T2 signal in MRI is closely related to water content (e.g. inflammatory oedema if higher or nucleus pulposus sclerosis if lower).
Response) We agree with the reviewer’s suggestion. We have added a statement that an increase in MRI T2 values within IVD suggests improvement of IVD degeneration (Page 6, lines 200 to 201, highlighted in gray)
5) Section 5.1, line 240: please explain better why the PRP-treated arm of patients was compared with a contrast agent-treated arm.
Response) Thank you for the reviewer’s comment. In the Tuakli-Wosornu study, a contrast agent was used to diagnose discogenic low back pain in provocative discography. Therefore, the contrast agent would be used in the control group.
6) Section 5.1: if available in the analyzed studies, objective comparison between PRP-treated patients and untreated controls by NMR or CT-scan should be reported and discussed.
Response) Thank you for the reviewer’s comment. Unfortunately, NMR or CT-scan has not been utilized to evaluate disc degeneration in the previously reported clinical studies of PRP for discogenic LBP patients.
Minor points
7) Page 2, section 2.1, line 69: please replace ’secretory organelle’ with ‘secretory inclusions’: usually, organelles are intended as the subcelluar structures needed for cell life.
Response) It was corrected accordingly.
8) Page 2, section 2.1, line 82, please rephrase as: ‘Platelets contain 3-8 dense granules, which contain high concentrations…’ indeed, dense granules are likely formed by the RER-Golgi secretory pathway and not by the endosomal-lysosomal pathway
Response) It was corrected accordingly.
9) Page 4, line 123, please spell out ‘PAW’; line 136, PPR should be PRP
Response) The description of PAW classification was revised as follows.
Page 4, lines 130 to 133:
DeLong et al. proposed a classification system based on Platelet concentration, Activation or not, and leukocyte (White blood cell) concentration (PAW classification), which can be used to quickly evaluate the PRP preparations used in multiple studies and clinical practice [25].
10) Page 5, line 207: please replace ‘neuronal regeneration’ with ‘axonal regeneration’. It is not believable that spinal cord or ganglionic neurons may actually regenerate upon PRP stimulation since they are a well-known post-mitotic cell population.
Response) It was corrected accordingly.
Round 2
Reviewer 2 Report
1. Line 124: Among the varies growth factors in PRP, why did the authors mentioned only PDGF-AB in this paragraph? Furthermore, the therapeutic effect of different platelet concentration should be described, just as the authors do in the dicussion of leukocyte content in the following paragraph.
2. Line 134: Some PRP preparations need activation while some need not. The authors should add further detailed description about the underlying mechanism how unactivated PRP works, and try to find literatures to compare the efficacy of activated and unactivated PRP in treatment.
3. Line 152: How does Pure Platelet-Rich Fibrin (P-PRF) prepared? Does this product has its water content removed from the preparation?
4. Figure 1 needs to be better organized. Decreased COX-2 is related to decreased inflammatory response, and decreased MMP-3 is related to increased collagen synthesis. Arrows are needed between the names mentioned above. Furthermore, the effect of Keap1-Nrf2 pathway should also be shown in the figure.
5. Line 513: The authors cannot draw conclusions about the effect of PRP because different studies have reported opposing results and the number of studies is insufficient. Actually, this can be solved by a formal meta-analysis to determine the odds ratio and the confidence interval of the pooled effect of current studies. The authors should at least explain why meta-analysis was not performed in this manuscript.
Author Response
Response to the reviewer’s comments
Reviewer 2
- Line 124: Among the varies growth factors in PRP, why did the authors mentioned only PDGF-AB in this paragraph? Furthermore, the therapeutic effect of different platelet concentration should be described, just as the authors do in the dicussion of leukocyte content in the following paragraph.
Response)
We appreciate the reviewer’s valuable comments. We added the following sentence regarding other growth factors and differences in platelet concentrations (highlighted yellow).
Page 4, lines 124 to 128:
Although PRP containing more than a specific concentration of platelets tends to have higher concentrations of platelet-derived growth factor-AB (PDGF-AB), the relation-ship was not always directly proportional. The concentrations of transforming growth factor beta-1 (TGF-β1) and vascular endothelial growth factor (VEGF) vary widely from system to system.
Page 4, lines 129 to 131:
A certain concentration of platelets is necessary because a low platelet concentration tends to reduce the number of growth factors.
- Line 134: Some PRP preparations need activation while some need not. The authors should add further detailed description about the underlying mechanism how unactivated PRP works, and try to find literatures to compare the efficacy of activated and unactivated PRP in treatment.
Response)
We appreciate the reviewer’s helpful comments. We added the following sentence detailing the PRP activation (highlighted yellow).
Page 4, lines 142 to 147:
In a systematic review and meta-analysis, activated PRP was reported to be more ef-fective for improving pain and functionality in patients with knee OA compared with non-activated PRP [27]. Additionally, Gentile reported that non-activated PRP was more useful for hair growth than activated PRP [28]. When PRP is injected into soft tissue, activation prior to administration is not always necessary because natural col-lagen type I acts as the activator [28].
- Line 152: How does Pure Platelet-Rich Fibrin (P-PRF) prepared? Does this product has its water content removed from the preparation?
Response)
As noted by the reviewer, we have included a detailed statement regarding the PRF (highlighted yellow).
Page 5, lines 142 to page 6, lines 171:
PRF is purified by collecting blood in dry glass or glass-coated plastic tubes and imme-diately centrifuging at a low RPM. PRF preparations have a high-density fibrin net-work, meaning they can be handled as if they are a solid material [36].
- Figure 1 needs to be better organized. Decreased COX-2 is related to decreased inflammatory response, and decreased MMP-3 is related to increased collagen synthesis. Arrows are needed between the names mentioned above. Furthermore, the effect of Keap1-Nrf2 pathway should also be shown in the figure.
Response)
We appreciate the reviewer’s comments. As the reviewer suggested, we have added the arrows and the effect of Keap1-Nrf2 pathway in the figure 1.
- Line 513: The authors cannot draw conclusions about the effect of PRP because different studies have reported opposing results and the number of studies is insufficient. Actually, this can be solved by a formal meta-analysis to determine the odds ratio and the confidence interval of the pooled effect of current studies. The authors should at least explain why meta-analysis was not performed in this manuscript.
Response)
We agree with the reviewer’s comments. A formal meta-analysis of current research is a powerful tool for determining the efficacy of any treatment for a specific disease or disorder. The primary aim of this article was to provide a comprehensive review of the current literature on basic mechanisms and emerging clinical applications for treatment of several spinal diseases with PRP. Therefore, in this review, we did not individually perform a meta-analysis for efficacy of PRP for intradiscal therapy, spinal fusion surgery, intraarticular therapy for the facet joint or sacroiliac joint pain, and epidural therapy. We have added the above description to the conclusion section (Page 17, lines 522 to 526: highlighted yellow).
Reviewer 3 Report
No additional comments to the revised version
Author Response
Thank you very much for reviewing our manuscript. The authors sincerely appreciate the reviewer's comment and understanding of this manuscript.